# Friendship habits questionnaire: A measure of group- versus dyadic-oriented socializing styles

**Philip Howlett**[1]*, **Gülseli Baysu**[1], **Anthony P. Atkinson**[2], **Tomas Jungert**[3],
**Magdalena Rychlowska**[1]

1 Queen's University Belfast, Belfast, United Kingdom, 2 Durham University, Durham, United Kingdom,
3 Lund University, Lund, Sweden

* phowlett01@qub.ac.uk

**Data Availability Statement:** The data files can be found under the sub-folders (e.g., Study 1, Study 2, Study 3). https://osf.io/98nrw/?view_only= e5a9be5c13d84aaaa1df9c786b531c38.

## Abstract

Friendships are central to our social lives, yet little is known about individual differences associated with the number of friends people enjoy spending time with. Here we present the Friendship Habits Questionnaire (FHQ), a new scale of group versus dyadic-oriented friendship styles. Three studies investigated the psychometric properties of group-oriented friendships and the relevant individual differences. The initially developed questionnaire measured individual differences in extraversion as well as desire for intimacy, competitiveness, and group identification, traits that previous research links with socializing in groups versus one-to-one friendships. In three validation studies involving more than 800 participants (353 men, age $M = 25.76$) and using principal and confirmatory factor analyses, we found that the structure of the FHQ is best described with four dimensions: extraversion, intimacy, positive group identification, and negative group identification. Therefore, competitiveness was dropped from the final version of the FHQ. Moreover, FHQ scores reliably predicted the size of friendship groups in which people enjoy socializing, suggesting good construct validity. Together, our results document individual differences in pursuing group versus dyadic-oriented friendships and provide a new tool for measuring such differences.

## Introduction

Friendship has been proposed as the most important aspect of human life influencing happiness and both physical and mental wellbeing [1]. This is a reasonable claim: friendship satisfies the needs of autonomy, competence, and relatedness throughout the lifespan [2] and friends are vital for socioemotional development during childhood and adolescence [3]. Children as young as five can describe and distinguish friendship groups from other groups, such as work groups [4], suggesting our early understanding of the important bond of friendship. However, not all of us make friends in the same way. Although humans generally value belonging to groups [5], not everybody wants to socialize in groups. Specifically, compared to men, women tend to socialize more in dyadic interactions [6–8]. Individual differences also matter, with extroverts and introverts socializing differently [9]. However, to our knowledge, there has

**Funding:** Author PH received a funded Studentship from the Department for the Economy, Northern Ireland (https://www.economy-ni.gov.uk/). The funders had no role in study design, data collection and analysis, decision to publish, or preparation of the manuscript.

**Competing interests:** The authors have declared that no competing interests exist.

been no attempt to examine which individual characteristics are associated with socializing in dyads versus groups. It is our aim to examine the individual differences associated with group versus dyadic-oriented socializing styles and to propose a novel tool for measuring such differences.

Current research on friendship often relies on friendship nominations, where participants name their friends [8, 10]. This method only provides information on *who* people interact with and not on whether these interactions are in groups or dyads. Moreover, existing questionnaires measuring friendship, such as the Friendship Qualities Scale [11], the McGill Friendship Questionnaire [12], and the Friendship Quality Questionnaire [13], tend to focus on friendship quality rather than on the number of friends with whom one enjoys socializing. There is little research on the size of groups that people encounter and spend time with, despite there being differences in the way that individuals socialize [14]. However, such investigations are needed because the number of others with whom people interact in their daily life has important implications not only for how close an individual can be with their friends [15], but also for the extent of their self-disclosure [8], and of their verbal and nonverbal communication [16, 17]. Moreover, the quantity of friends contributes to life satisfaction and wellbeing [18, 19]. Despite the importance of studying the number of friends with whom people enjoy interacting, it is hard to measure naturally occurring friendship interactions and, while possible, conducting laboratory studies with pairs and groups of friends are methodologically and organizationally challenging. Furthermore, real-time observations of friendship groups are intrusive and estimating the size of our social world via self-report tends to be unreliable, pointing to the need for creative and indirect ways of measuring the size of friendship groups. Accordingly, a literature search of the terms "dyads", "groups", and "differences" shows that extant studies have identified trait differences between people who socialize in groups versus dyads [6–9]. Here we present the Friendship Habits Questionnaire (FHQ), a new tool measuring whether one's socializing style is more group versus dyadic-oriented based on individual differences in extraversion, group identification, need for intimacy and competitiveness. Below we describe the dimensions of this questionnaire and present three studies testing its factor structure.

*Extraversion* is the first construct we examine in the context of group- versus dyadic-oriented friendship styles. Even though all the Big Five personality traits are relevant to social networks and friendship [10], extraversion has the strongest link to network size [e.g., 20]. Compared to introverts, extraverts are more inclined to nominate others as friends [21, 22]. Their social networks are also more peer- versus family-oriented [23] and larger, especially regarding closest friends and family [20, 24]. Finally, extraverts have a greater tendency to define themselves in terms of group membership and their friends are more likely to be connected to one another [9]. In sum, extraverts have larger, denser, and more interconnected social networks [10], which suggests they might also socialize in groups rather than in dyads.

*Group identification* is another potentially relevant dimension of friendship orientation. Group identification is linked to the concepts of relational versus collective self-construals, associated with defining oneself in terms of important relationships or in terms of group membership, respectively [25]. Relational self-construal overlaps with closeness and intimacy [26, 27] and collective self-construal is closely linked with group identification, or the extent to which group membership affects one's emotion, self-esteem, and identity [28, 29]. Reciprocated friendship in groups also leads to greater group identification [30]. Here we predict that individuals with more collective self-construals who strongly identify with their friendship groups, will enjoy spending time with more people.

Another relevant construct is *intimacy*, which we define here as disclosing personal information and mutual support [31]. Self-disclosure tends to be higher in dyads than in groups

[32]. Research on gender also links intimacy with dyadic-oriented friendships, because women tend to have more dyadic interactions whereas men tend to socialize in groups, a difference observed in different cultures and species [33, 34] and replicated in the context of virtual interactions [6]. One potential explanation is that, compared to men, women put greater importance on self-disclosure and intimacy [e.g., 31]. Consistent with such interpretations, studies find that men choose having more friends over few intimate friendships whereas women sacrifice quantity of friends for higher intimacy [6, 8]. Based on these findings, we predict that, compared to group-oriented individuals, dyadic-oriented individuals should attach greater value to friendship intimacy, self-disclosure, and mutual support.

We also expect group-oriented individuals to be more competitive than their dyad-oriented peers, with *competitiveness* being defined as the desire to win [35, 36] and enjoyment of competition and contentiousness [35]. This prediction is also motivated by the literature on gender: men's friendships involve competing for a good position within a group to a greater extent than women's friendships [25] and teenage boys tend to value school peers' performance in school and sport more than girls [37]. Overall, men participate in more competitive interactions, potentially facilitating the emergence of friendship hierarchies [34]. Competitiveness is also more prevalent in groups than in dyads [e.g., 38], suggesting a connection between group friendships and a competitive tendency.

Bringing these different dimensions together, the current research aims to examine individual differences associated with socializing in groups versus dyads with a novel questionnaire that is proposed to measure the likelihood of socializing in groups of friends. Based on previous findings [20–38], we expect dyadic-oriented people to be more introverted [20–24], to identify less with friendship groups [25–30], to have a higher need for friendship intimacy [31–34] and to be less competitive than group-oriented individuals [35–38]. To measure these traits and to quantify group- versus dyadic-oriented friendship styles, we have created the Friendship Habits Questionnaire (FHQ), where high scores indicate that a person is more likely to socialize in friendship groups and low scores indicate that a person is more likely to socialize in dyads.

The FHQ comprises 30 statements reflecting the dimensions of extraversion, group identification, intimacy, and competitiveness (see Table 1). Items were adapted from existing questionnaires. Extraversion was measured with the 8 items (E1-E8, Table 1) of the Big Five Inventory [39]. We adapted items to describe friendship groups, e.g., *I am talkative* was changed to: *I am talkative when I am in a larger group of friends.*

The 9 items measuring group identification (GP1-GP5 and GN1-GN4, Table 1) were taken from the Group Identification Scale [29]. Items were modified to describe friendship groups, e.g., *I feel strong ties to this group* was changed to: *I feel strong ties to a friendship group.*

The need for intimacy was assessed with the subscale of Intimate Exchange in the Friendship Quality Questionnaire [13] (I1-I6, Table 1). Statements serving to describe specific relationships (e.g., *Jamie and I tell each other secrets*) were adapted to describe friends in general (e.g., *My friends and I tell each other secrets*).

The seven items measuring competitiveness (C1-C7, Table 1) were adapted from the Revised Competitiveness Index [35]. We selected 4 items measuring enjoyment of competition (e.g., *I enjoy competing against an opponent*) and 3 items measuring contentiousness (e.g., *I try to avoid arguments*, reverse-scored) with the highest component loadings and we adapted them to describe friendships (e.g., *I enjoy competing against a friend* and *I try to avoid arguments with friends*). Fig 1 shows the research process of refining the FHQ across the three studies. In Study 1, graduate social science and business students evaluated the items of the FHQ over an 8-day period from December 2019 to January 2020. We used these ratings to explore the questionnaire's component structure and psychometric properties [40, 41]. We followed a

**Table 1. Study 1: Descriptive statistics and component loadings for the PCA analysis with oblimin rotation.**

| Dimension | FHQ Items | M | SD | Component Loading | | | | | |
|---|---|---|---|---|---|---|---|---|---|
| | | | | 1 | 2 | 3 | 4 | 5 | 6 |
| | | | | EXTR | INTIM | COMP | POS GROUP ID | NEG GROUP ID | CONT |
| Extraversion (M = 3.02, SD = .41) | E1 I am outgoing and sociable when I am in a larger group of friends | 3.10 | 1.01 | **.81** | | | | | |
| | E2 I am talkative when I am in larger group of friends | 3.04 | 1.14 | **.81** | | | | | |
| | E3 I am reserved when I am in larger group of friends$^\Delta$ | 2.96 | 1.17 | **-.39** | | | | .39 | |
| | E4 I am full of energy when I am in larger group of friends | 2.96 | 1.08 | **.66** | | | | | |
| | E5 I tend to be quiet when I am in larger group of friends$^\Delta$ | 2.98 | 1.16 | **-.43** | | | | | |
| | E6 I have an assertive personality | 3.04 | 1.00 | **.62** | | | | | -.32 |
| | E7 I am sometimes shy and inhibited in larger group of friends$^\Delta$ | 3.08 | 1.15 | **-.32** | | | | | |
| | E8 I generate a lot of enthusiasm when I am in larger group of friends | 3.00 | 1.10 | **.84** | | | | | |
| Intimacy (M = 3.53***, SD = .78) | I1 My friends and I always tell each other our problems$^\Delta$ | 3.51 | 1.06 | | **.71** | | .30 | | |
| | I2 My friends and I talk about the things that make us sad$^\Delta$ | 3.49 | 1.06 | | **.79** | | | | |
| | I3 I tell my friends when I am mad about something that happened to me$^\Delta$ | 3.67 | 0.90 | | **.52** | | | | -.38 |
| | I4 My friends and I tell each other secrets$^\Delta$ | 3.67 | 1.01 | | **.73** | | | | |
| | I5 My friends and I tell each other private things$^\Delta$ | 3.98 | 0.83 | | **.79** | | | | |
| | I6 My friends and I talk about how to make ourselves feel better if we are mad at each other$^\Delta$ | 2.86 | 1.17 | .37 | **.63** | | | | |
| Competitiveness (Enjoyment of Competition) (M = 2.81$^\dagger$, SD = .66) | C1 I like competition among friends | 2.31 | 1.28 | .30 | | **.72** | | | |
| | C2 I enjoy competing against a friend | 2.55 | 1.19 | | | **.83** | | | |
| | C3 I don't like competing against a friend$^\Delta$ | 3.49 | 1.29 | | | **-.82** | | .47 | |
| | C4 I am a competitive individual | 2.90 | 1.25 | | | **.67** | | .41 | |
| Competitiveness (Contentiousness) (M = 3.19, SD = .82) | C5 I will do almost anything to avoid an argument with friends$^\Delta$ | 3.14 | 1.08 | | | | | | **.92** |
| | C6 I try to avoid arguments with friends$^\Delta$ | 3.37 | 1.04 | | | | | | **.84** |
| | C7 I often remain quiet rather than risk hurting another friend's feelings$^\Delta$ | 3.06 | 0.88 | -.42 | .41 | .40 | | | **.45** |
| Group Identification (Positive) (M = 3.47***, SD = .84) | GP1 I am glad when I belong to a friendship group | 3.61 | 1.02 | | | | | | |
| | GP2 I identify with a friendship group | 3.45 | 1.14 | | | | **.89** | | |
| | GP3 I feel strong ties to a friendship group | 3.39 | 1.15 | | .30 | | **.66** | | |
| | GP4 I think friendship groups work well together | 3.49 | 0.85 | | | | **.73** | | |
| | GP5 I see myself as an important part of a friendship group | 3.39 | 1.10 | | | | **.63** | | |

*(Continued)*

**Table 1.** (Continued)

| Dimension | FHQ Items | M | SD | Component Loading | | | | | |
|---|---|---|---|---|---|---|---|---|---|
| | | | | 1 | 2 | 3 | 4 | 5 | 6 |
| | | | | EXTR | INTIM | COMP | POS GROUP ID | NEG GROUP ID | CONT |
| Group Identification (Negative) (M = 2.40***, SD = .92) | GN1 I feel held back in friendship groups△ | 2.39 | 1.08 | | | | - | .52 | |
| | GN2 I do not consider a friendship group to be important△ | 2.31 | 1.16 | | | .31 | | .46 | |
| | GN3 I do not fit in well with other members of friendship groups△ | 2.51 | 1.18 | | | | | .82 | |
| | GN4 I feel uneasy with members of friendship groups△ | 2.41 | 1.14 | | | | | .70 | |

Note. EXTR = Extraversion, INTIM = Intimacy, COMP = Competitiveness, POS GROUP ID = Positive Group Identification, NEG GROUP ID = Negative Group Identification, CONT = Contentiousness. Component loadings smaller than .30 are omitted and highest component loadings for each item are in boldface.

△s denote reverse-scored FHQ items in Column 2, but please note that intimacy scores were unreversed in the final version of the FHQ (see Study 2 and 3 for clarifications).

Asterisks in Column 1 denote values significantly different from 3, the scale midpoint,

† p < .10,

*p < .05,

**p < .01, and

***p < .001.

Differences were tested using one-sample t-tests, except for Extraversion. The distribution of this component departed from normality, $W(48) = .93$, $p < .008$ and we used a One-Sample Wilcoxon Signed Rank Test.

procedure similar to the approach recommended by Hinkin and Tracey [42] who argue that providing participants with theoretically motivated definitions of the underlying construct reduces subjectivity. These authors also argue that a theoretical approach is as effective in designing high-quality questionnaires as an item-reduction approach that involves participants completing an exhaustive list of items assessing many related dimensions. Therefore, the goal of this approach is to test if participants respond similarly to items indexing specific dimensions of a theoretically defined underlying construct.

In Study 2, we administered the FHQ over a 4-week period in January 2020 to a new, larger participant sample and further investigated its structure using confirmatory factor analyses. We also assessed the construct validity of the questionnaire by testing whether high FHQ scores are associated with socializing in larger groups, and thus low scores are associated with socializing in dyads.

Finally, in Study 3, we re-examined the FHQ factor structure in a new sample, collected over a 10-week period from February 2020 to April 2020, using the best-fitting model determined in Study 2. We also tested whether participants' FHQ scores were associated with their socializing preferences and practices.

## Study 1

### Aims and hypotheses

Study 1 aimed to evaluate the items of the FHQ and tested their relevance to the theoretical dimensions studied. We predicted that the FHQ items would be rated as relevant to friendship practices and that items would be associated with four components corresponding to extraversion, group identification, intimacy and competitiveness.

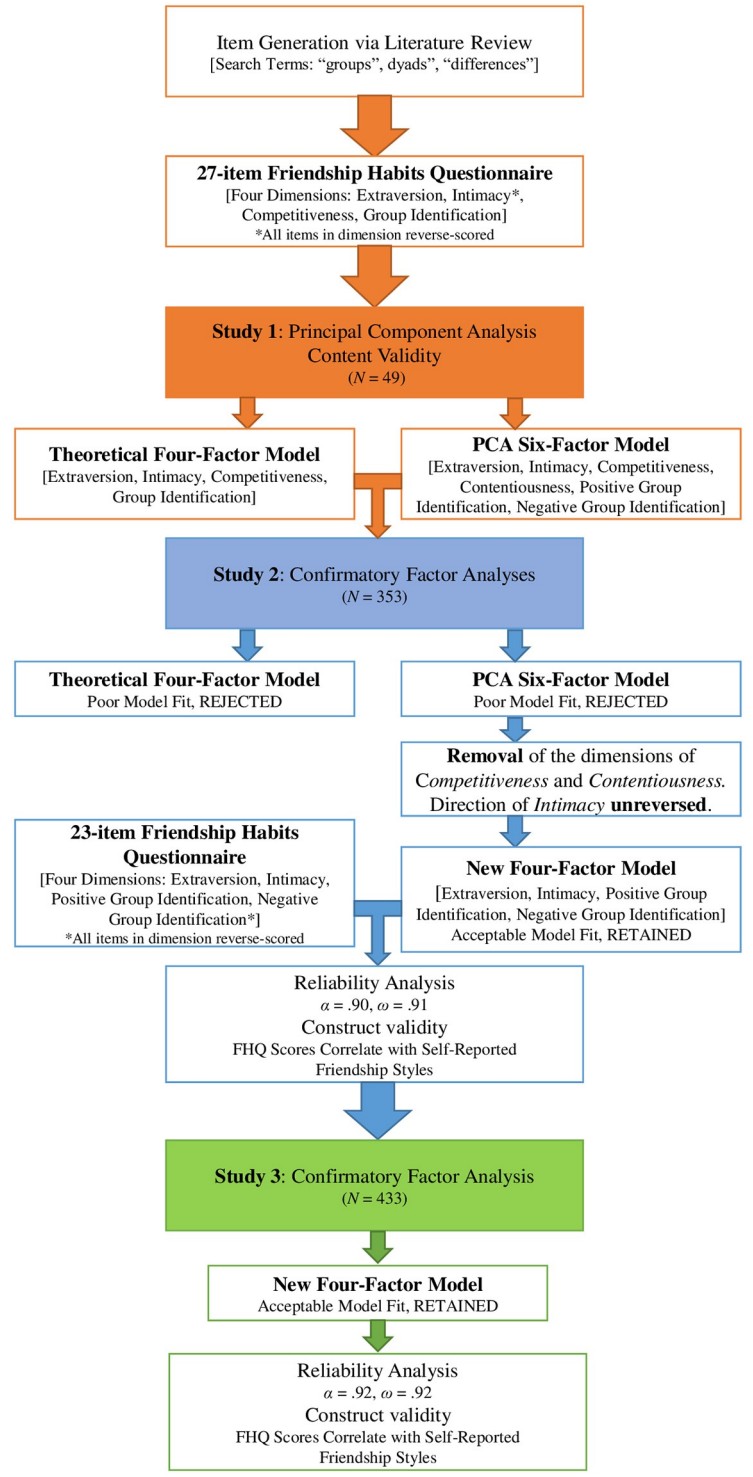

**Fig 1. Research process and main goals of each study in the validation of the Friendship Habits Questionnaire (FHQ).**

## Methods

**Participants.** Forty-nine volunteers (22 male, 2 unknown, age $M = 26.04$, $SD = 6.10$) completed the survey, in line with earlier research on questionnaire validation [42]. Participants were recruited on survey-swapping Facebook groups for graduate students in social sciences and business.

**Procedure and materials.** The survey was presented online using Qualtrics [43]. After providing consent, participants read the definition of friendship habits, defined as the way in which people socialize, with dyadic-oriented people described as being introverted, not identifying with a friendship group, enjoying close and intimate friendships, and disliking competition among friends. Group-focused people were described as extraverted, identifying with a friendship group, preferring group interactions and less intimate friendships, and enjoying competition among friends.

The FHQ items were presented in a random order on the same page. Participants used 5-point scales ranging from *Not at all* to *Completely* to rate how much each statement described friendship habits. Afterwards, subjects provided demographic information, were thanked and debriefed.

**Ethics.** An Ethics Committee at the Department of Psychology, Lund University, has corroborated that the present research protocol follows the research ethics guidelines that must be followed in Sweden for all three studies in this research. For each study, participants read an informed consent page within the online surveys and showed their consent by clicking an arrow button to proceed.

## Analytic strategy

We analyzed the data using the 'jamovi' and 'lavaan' packages and Rstudio (version 1.4.1717). Ratings were reversed when necessary (see Δs in Table 1) and entered in a PCA using oblimin rotation with Kaiser normalization, as we expected our factors to be correlated. However, to test the robustness of the findings, we also ran a PCA analysis using varimax rotation, following the procedure recommended by Hinkin and Tracey [42].

## Results

The analysis revealed eight components with eigenvalues higher than 1.00 [44], explaining 79.44% of the variance. The variance explained levelled off after four components (S1 Fig), but these components failed to explain a minimum of 60% of the variance [45, 46]. We therefore inspected all eight components.

As seen in Table 1, the first component, Extraversion (15.53% variance explained), was reflected by the eight items measuring this dimension (E1-E8). The same was true for the second component, intimacy (12.80% variance explained), loading on the six intimacy items (I1-I6). The third component, competitiveness (10.40% variance explained), corresponded to the four items measuring the enjoyment of competition (C1-C4). The fourth component (10.86% variance explained) was reflected by four of the nine items measuring group identification (GP2-GP5) and was named positive group identification. The fifth component (9.92% variance explained) corresponded to four reverse-scored items measuring group identification (GN1-GN4) and was named negative group identification. Finally, the sixth component, contentiousness (7.89% variance explained), loaded on the three items measuring contentiousness (C5-C7). The two remaining components were not interpretable and were excluded from further analyses (One component loaded on five items: E3, E5, E7, GP5 and, negatively, on I5, see Table 1. The second component loaded on three items, GP1 and, negatively, on I6 and GN2, see Table 1). The six retained components explained 67.40% of the variance and could be

linked to the dimensions of the FHQ. All items except two (E3 and E7) had component loadings higher than .40 [45, 46] (see Table 1). Table 1 also includes the descriptive statistics of relevance ratings for the six components. The only item that did not load onto one of these six components was GP1: *I am glad when I belong to a friendship group*. However, since participants rated GP1 as the most relevant of all group identification items (see Table 1), we decided to retain this item for further exploration. Moreover, the supplemental analysis with varimax rotation, which showed a similar pattern of six interpretable components (see S2 Fig, S1 Table), supported our decision to retain this item as it was associated with positive group identification as expected.

## Discussion

Study 1 examined the component structure of the FHQ. Instead of the four predicted dimensions, the PCA revealed six components: extraversion, intimacy, competitiveness, positive group identification, negative group identification, and contentiousness.

Although surprising at the first sight, the emergence of two additional components can be explained by the scales that informed the FHQ. The components of competitiveness (C1-C4, Table 1) and contentiousness (C5-C7) reflect the two facets of competitiveness measured by the Revised Competitiveness Index [35]. Interestingly, items measuring group identification loaded on two components: positive group identification (GP1-GP5) and negative group identification (GN1-GN5). This finding partly replicates the PCA of the Group Identification Scale [29], from which we borrowed the corresponding FHQ items. Our component of positive group identification overlaps with the subscale of Emotional Aspects of Group Identification [29]. In addition to the four items of this subscale, positive group identification was associated with another positively worded item: *I feel strong ties to a friendship group*, linked with the Cognitive subscale [29]. In our analysis, the reverse-scored items GN1-GN5 reflected a single component of negative group identification. In the research by Hinkle and colleagues, three of these questions belonged to two different subscales and one was uncategorized [29]. It is thus possible that items indexing less-well defined components were grouped into one single dimension in the present research.

The inspection of relevance ratings (see Table 1, column 1) revealed that, although items measuring intimacy and positive group identification were perceived as relevant to friendship habits, ratings of extraversion and contentiousness were not different from the midpoint of the scale [i.e., 3], and ratings of competitiveness and negative group identification were lower than the midpoint of the scale. Although the last result may seem surprising, these two components capture emotions and tendencies potentially perceived as undesirable, such as enjoying competition with friends or having negative feelings about friendship groups. It is therefore possible that participants rated these items as less relevant because the corresponding behaviors and attitudes were perceived more negatively. Given the small sample size, Study 2 explored the factor structure of the FHQ using a larger participant pool.

## Study 2

### Aims and hypotheses

Study 2 aimed to test if the structure of the FHQ is best explained by the four theoretical dimensions (extraversion, group identification, intimacy, competitiveness) that we initially expected or by two alternative models, including the six-factor model that emerged from Study 1. Another aim of the study was to establish reliability and construct validity of the FHQ by comparing it self-reported measures of friendship preferences and practices. We predicted that a six-factor model of FHQ will have a more acceptable fit than the theoretically motivated

four-factor model. We also hypothesized that higher FHQ scores will be associated with larger self-reported friendship group sizes and a greater preference to be in larger friendship groups.

## Methods

**Participants.** Three hundred fifty-three volunteers (162 male, 1 other, 5 unknown, age $M = 26.39$, $SD = 7.75$) completed the study. They were recruited on survey-swapping Facebook groups and via SurveyCircle [47], an online research community. No participant was excluded. Although we didn't run a power analysis, we aimed to recruit a minimum of 150 participants, a sample size deemed sufficient for a CFA [48].

**Procedure and materials.** After providing consent, participants completed the FHQ, rating their agreement with each statement on 7-point scales ranging from *Strongly disagree* to *Strongly agree*. They then provided demographic information and answered questions about their friendship preferences and practices. Participants were first asked about the number of people that they usually socialize with, and selected one of three response options: *one-to-one*, *2–3 people excluding yourself*, or *4+ people excluding yourself*. Participants also reported the ideal number of friends that they like to socialize with at the same time, using free response format. Finally, subjects were instructed to think about the last 3 months and used a free response format to report the number of friends they usually socialized with, the number of friends present in the interaction(s) in which participants felt most comfortable, and the number of friends present in the most enjoyable interaction(s). Participants also reported the number of times they socialized with one friend, two or three friends, and four or more friends. After completing the questionnaire, subjects were thanked and debriefed.

## Analytic strategy

While PCA is suitable for initial data screening, CFA is more appropriate for explaining relationships between variables and testing how well the structure of the data fits a specific model [40, 46, 49]. In Study 2, we used the 'jamovi' and 'lavaan' packages and Rstudio (version 1.4.1717) and conducted a series of second-order CFAs to test whether the factor structure of the Friendship Habits Questionnaire shows a better fit with the theory-driven four-factor model or with the six-factor model suggested by the results of Study 1. In the four-factor model, friendship styles are comprised of four first-order dimensions (extraversion, group identification, intimacy, competitiveness), each indexed by the items measuring these constructs. In the six-factor model friendship styles involve six first-order dimensions (intimacy, competitiveness, extraversion, positive group identification, negative group identification, contentiousness), indexed by the items linked with these components in Study 1. In each case, all first-order factors load on one second-order factor (i.e., friendship styles). After assessing the fit of these models, we explored an alternative model. To allow for non-normal distribution in our data, the CFAs used the maximum likelihood estimator with robust standard errors (MLR). Model fit was assessed using the robust relative chi-square, $\chi^2/\mathrm{df} < 2.0$ [40], the robust comparative fit index (CFI), the robust Tucker-Lewis index (TLI; for both, $>.90$ –acceptable, $>.95$ –excellent) [50], the standardized root mean square residual (SRMR $< .08$) [51], and the robust root mean square error of approximation (RMSEA $< .06$) [51]. Finally, we inspected the reliability of the FHQ and examined the correlations between FHQ scores, friendship preferences and practices, and self-reported friendship group sizes.

## Results

**Confirmatory factor analyses.** Table 2 presents fit indices for the four-factor and the six-factor model (for factor loadings, see S2 and S4 Tables respectively). Both models showed a

**Table 2. Study 2: Model fit indices for the four-factor theoretical model and the six-factor model derived from study 1.**

| Model | $\chi^2$ | df | $\chi^2$/df | CFI | TLI | SRMR | RMSEA | RMSEA 90% CI |
|---|---|---|---|---|---|---|---|---|
| 4-Factor Model | 1078.81*** | 401 | 2.69 | .852 | .839 | .089 | .074 | (.068, .079) |
| 6-Factor Model | 848.74*** | 399 | 2.13 | .902 | .893 | .083 | .060 | (.054, .065) |

***p < .001.

poor fit across all indices. Factor correlations of the six-factor model (see Table 3) revealed that the two dimensions of competitiveness, that is, contentiousness and enjoyment of competition, were only weakly correlated with other dimensions. Competitiveness was also weakly correlated with other dimensions in the four-factor model (S3 Table). Unexpectedly, low desire for intimacy was negatively correlated with other variables (note that this dimension was reverse-scored such that higher scores reflect *lower* need for intimacy, see Table 1).

We therefore removed the dimensions of contentiousness and competitiveness and created a new four-factor second-order CFA model including only first-order dimensions of intimacy, extraversion, positive group identification, and negative group identification. In this new model, intimacy was also unreversed to reflect its positive correlations with other FHQ variables. As shown in Table 4, this new model showed an acceptable fit on all indices and was retained for further analyses. All the factor items loaded significantly on their respective first-order factors and these factors in turn loaded significantly on the second-order friendship styles factor (all standardized loadings > .40). Table 5 shows factor correlations.

**Reliability.** We examined reliability using Cronbach's alpha and McDonald's omega. Extraversion subscale had excellent reliability ($\alpha$ = .91, $\omega$ = .91), subscales of intimacy and positive group identification had good reliability ($\alpha$ = .83, $\omega$ = .84 and $\alpha$ = .83, $\omega$ = .84, respectively), and negative group identification subscale had acceptable reliability ($\alpha$ = .69, $\omega$ = .70). The entire FHQ had excellent reliability ($\alpha$ = .90, $\omega$ = .91).

**Scoring the friendship habits questionnaire.** To calculate the FHQ scores, we first computed reverse scores for items marked with Δs in Table 1 (except those indexing intimacy, which were left unreversed). Responses to the 23 items (without contentiousness and enjoyment of competition) were then averaged for each participant. These averages range from 1 to 7, with higher values indicating group-oriented friendship styles and lower scores indicating dyad-oriented friendship styles.

**Construct validity.** We next examined the relationships between FHQ scores and self-reported friendship preferences and practices. Participants reported the number of people they

**Table 3. Study 2: Factor correlations for the six-factor model.**

| Factor | 1 | 2 | 3 | 4 | 5 | 6 |
|---|---|---|---|---|---|---|
| 1. Extraversion | - | | | | | |
| 2. Enjoyment of Competition | .16** | - | | | | |
| 3. Contentiousness$^\Delta$ | .24*** | .33*** | - | | | |
| 4. Intimacy$^\Delta$ | -.27*** | -.01 | -.08 | - | | |
| 5. Positive Group Identification | .51*** | .21*** | -.04 | -.49*** | - | |
| 6. Negative Group Identification$^\Delta$ | .61*** | .01 | .21** | -.40*** | .73*** | - |

Note.

$^\Delta$s denote reverse-scored FHQ dimensions.

Intimacy scores were unreversed in the final version of the FHQ, see Study 2 and 3 for clarifications.

**Table 4. Study 2: Model fit indices and factor loadings for the new four-factor model (intimacy, extraversion, positive group identification, negative group identification).**

| | $X^2$ | Df | $\chi^2$/df | CFI | TLI | SRMR | RMSEA | RMSEA 90% CI |
|---|---|---|---|---|---|---|---|---|
| New 4-Factor Model | 423.92*** | 226 | 1.88 | .939 | .932 | .056 | .054 | (.046, .062) |

| Factor | Item | Estimate | Completely Standardized Solution | SE | $p$ |
|---|---|---|---|---|---|
| Extraversion | E1 | 1.14 | .85 | .08 | < .001 |
| | E2 | 1.17 | .87 | .07 | < .001 |
| | E3 | 1.11 | .83 | .07 | < .001 |
| | E4 | 1.02 | .79 | .08 | < .001 |
| | E5 | 1.16 | .82 | .08 | < .001 |
| | E6 | .54 | .42 | .08 | < .001 |
| | E7 | .92 | .67 | .08 | < .001 |
| | E8 | .88 | .73 | .07 | < .001 |
| Intimacy | I1 | .96 | .75 | .08 | < .001 |
| | I2 | .88 | .64 | .08 | < .001 |
| | I3 | .69 | .58 | .08 | < .001 |
| | I4 | 1.07 | .83 | .07 | < .001 |
| | I5 | .98 | .80 | .08 | < .001 |
| | I6 | .64 | .44 | .08 | < .001 |
| Positive Group Identification | GP1 | .41 | .62 | .08 | < .001 |
| | GP2 | .66 | .78 | .11 | < .001 |
| | GP3 | .66 | .81 | .11 | < .001 |
| | GP4 | .39 | .62 | .07 | < .001 |
| | GP5 | .59 | .71 | .09 | < .001 |
| Negative Group Identification | GN1 | .44 | .61 | .11 | < .001 |
| | GN2 | .42 | .53 | .10 | < .001 |
| | GN3 | .50 | .71 | .13 | < .001 |
| | GN4 | .41 | .60 | .11 | < .001 |
| Friendship Styles | Extraversion | .81 | .63 | .11 | < .001 |
| | Intimacy | .59 | .51 | .09 | < .001 |
| | Positive Group ID | 1.62 | .85 | .34 | < .001 |
| | Negative Group ID | 1.87 | .88 | .53 | .001 |

***p < .001.

**Table 5. Study 2: Factor correlations for the new four-factor model.**

| Factor | 1 | 2 | 3 | 4 |
|---|---|---|---|---|
| 1. Extraversion | - | | | |
| 2. Intimacy | .26*** | - | | |
| 3. Positive Group Identification | .52*** | .49*** | - | |
| 4. Negative Group Identification$^\Delta$ | .60*** | .40*** | .74*** | - |

Note.
$^\Delta$s denote reverse-scored FHQ dimensions.

**Table 6. Study 2: Descriptive statistics and spearman correlations between FHQ scores (23 items) and measures of friendship practices.**

| Variable | n | M | SD | Min | Max | 1 | 2 | 3 | 4 | 5 | 6 | 7 |
|---|---|---|---|---|---|---|---|---|---|---|---|---|
| 1. FHQ | 353 | 4.81 | .89 | 1.00 | 7.00 | - | | | | | | |
| 2. Ideal Number of Friends | 339 | 3.69 | 5.69 | 1.00 | 100.00 | .36*** | - | | | | | |
| 3. Usual Number of Friends | 347 | 3.69 | 2.84 | 0.00 | 30.00 | .20*** | .51*** | - | | | | |
| 4. Comfortable Number of Friends | 346 | 3.13 | 2.19 | 0.00 | 20.00 | .26*** | .61*** | .62*** | - | | | |
| 5. Enjoyable Number of Friends | 346 | 3.62 | 2.49 | 0.00 | 20.00 | .32*** | .57*** | .47*** | .69*** | - | | |
| 6. Average Group Size | 336 | 2.17 | .58 | 1.00 | 4.00 | .16** | .27*** | .42*** | .40*** | .38*** | - | |
| 7. Proportion of Time in Groups | 336 | .56 | .23 | 0.00 | 1.00 | .13* | .23*** | .37*** | .36*** | .32*** | .96*** | - |

usually spent time with and the ideal number of friends to socialize with at the same time. They also listed, for the last 3 months, the number of friends that they usually socialized with and the number of friends present in the most comfortable and most enjoyable interactions. Additionally, we used participants' reports of the number of times they socialized with one friend, two or three friends, or four or more friends during the last 3 months, to compute two new variables. The average group size was calculated by multiplying the frequency of interactions by 1 (one-to-one), 2.5 (two or three friends), and 4 (four or more friends), adding the scores, and dividing the sum by the total number of interactions reported by each participant. The proportion of time in groups was calculated by adding the interactions with two or three friends and four or more friends and dividing the sum by the total number of interactions. Table 6 displays correlations between FHQ scores and measures of friendship practices. FHQ scores were consistently and positively associated with self-reports of enjoying group interactions. FHQ scores also varied depending on participants' categorical responses describing the number of friends they usually socialized with, $F(2, 344) = 28.98$, $p < .001$, $\eta^2 p = .14$. Tukey post-hoc tests suggested that participants who socialized one-to-one had significantly lower FHQ scores ($M = 4.24$, $SD = .94$) than both participants who socialized in groups of 2–3 ($M = 4.78$, $SD = .83$), $p < .001$, and those that socialized in groups of 4 or more ($M = 5.30$, $SD = .75$), $p < .001$. Participants who socialized in groups of 2–3 scored less than those that socialized in larger groups, $p < .001$. Most participants ($n = 204$, 58.79%) reported socializing in groups of 2–3, compared to 61 (17.58%) people reporting more socialization in one-to-one settings and 82 (23.63%) people more time in groups of 4+.

Additional analyses examined the relationship between FHQ scores and the theoretically unrelated participants' level of education ($M = 3.76$, $SD = 1.00$), measured on a 5-point scale ranging from 1 (no formal education) to 5 (PhD/Doctorate degree). This correlation was not significant, $r_S (330) = .08$, $p = .128$.

## Discussion

Study 2 further examined the structure of the FHQ and explored the associations between FHQ scores and self-reported measures of enjoying group versus dyadic interactions. Confirmatory factor analyses showed that neither the theoretically motivated four-factor model of friendship styles nor the six-factor model derived from Study 1 were satisfactory in explaining variance in the data. The best fit was a modified four-factor model including extraversion, intimacy, positive group identification, and negative group identification. We therefore removed items measuring contentiousness and competitiveness. Further analyses showed that the entire scale and each component had good to excellent interitem reliability. Finally, FHQ scores covaried with how much participants enjoyed group versus dyadic interactions. Thus, people's

socializing habits could be reliably measured by assessing their extraversion as well as the desire for intimacy and group identification.

One counterintuitive finding was that intimacy, initially expected to be associated with dyadic friendships, was positively correlated with FHQ scores, such that respondents who valued intimacy had more group-oriented friendship styles. This is perhaps not surprising as intimacy is highly valued in friendships [31]. Intimacy and self-disclosure between friends are also common among extraverts [52, 53]. Therefore, it might be that increased sociability alone, often related to extraversion, increases intimacy. Although a previous study [32] showed that self-disclosure occurs more in dyads than in larger groups, the authors of this past study focused on interactions between strangers, and it is unclear how self-disclosure would play out among friends. Furthermore, people often move between dyads and groups [54], and relatively little is known about how group size impacts self-disclosure and intimacy.

FHQ scores were more strongly associated with participants' socializing preferences (i.e., ideal, most comfortable, and most enjoyable number of friends present) than with self-reported behaviors (number of people usually met at the same time, average group size, proportion of time in groups). Participants' free responses also point to the ambiguity of the concept of friendship. For example, some respondents were unsure if partners, work colleagues, or sports teams were friends or stated that their ideal number of people to socialize with depended on the activity. In Study 3, we aimed to reduce such ambiguities by providing a description of friendship interactions with a list of example activities.

## Study 3

### Aims and hypotheses

Study 3 aimed to retest the validity of the four-factor structure of the FHQ which emerged as the best-fitting model from Study 2 (i.e., extraversion, intimacy, positive group identification and negative identification) in a new sample sufficient for testing model fit [55]. Another aim was to further support the reliability and construct validity of the FHQ by comparing it with self-reported measures of friendship preferences and practices. We predicted that the four-factor model from Study 2 would have an acceptable fit and that high FHQ scores would be associated with larger self-reported friendship group sizes as well as a greater preference to be in larger friendship groups.

### Methods

**Participants.** Five hundred eighty-four volunteers were recruited on a large university campus and via survey-swapping Facebook groups, Reddit forums, and SurveyCircle [47]. We analyzed the data from participants who responded to all FHQ items and had correctly passed an attention check, excluding 151 responses, for a total sample of 433 participants (169 male, 5 other, age $M$ = 24.86, $SD$ = 5.48).

**Procedure and materials.** Participants completed the FHQ items, which were presented on the same page in a random order and included one attention check (*Please choose the option labelled strongly disagree*). We have added the attention check to ensure the highest possible quality of responses, as the entire questionnaire took approximately 15 minutes to complete and some participants might have felt fatigued completing it. Next, participants provided demographic information and answered questions about their friendship preferences and practices in general and in the last 3 months. These were mostly identical to the questions asked in Study 2, with minor wording changes and more specific descriptions. Specifically, to provide additional clarifications compared to Study 2, before answering questions about self-reported friendship interactions, participants were told that friendship interactions were times

when they agreed to meet for an activity with one or more of their friends. Examples of activities included *"bowling, watching a film at home or the cinema, chatting over tea or coffee, playing sports/video games/board games, going to a party, eating lunch/dinner together, going to a pub, going clubbing. (This list is not exhaustive)"*. Moreover, in Study 3, participants reported the number of people they usually socialized with by selecting among four options: *one-to-one, 2 people excluding myself, 3 people excluding myself, 4+ people excluding myself*. This was to reflect any potential differences between friendship groups with 2 other people and friendship groups with 3 other people. A second attention check asked participants to pick which time frame the questions referenced (3 months). After completing the questionnaire, participants were thanked and debriefed.

### Analytic strategy

Applying the same assessment criteria as Study 2, we conducted a CFA using the maximum likelihood estimator with robust standard errors (MLR) for the new four-factor model (extraversion, intimacy, positive group identification and negative identification). Again, similar to Study 2, we inspected the reliability of the FHQ and its correlations with other measures of friendship preferences and sizes of friendship groups.

**Confirmatory factor analysis.** We assessed the goodness of fit of the four-factor second-order model using the same criteria as in Study 2. Table 7 shows fit indices. The model showed an acceptable fit with the exception of the relative chi-square and RMSEA (see Tables 7 and 8 for factor loadings and correlations).

**Reliability.** Reliability was excellent for the extraversion subscale ($\alpha$ = .92, $\omega$ = .92), good for intimacy ($\alpha$ = .85, $\omega$ = .86) and positive group identification ($\alpha$ = .84, $\omega$ = .85), and acceptable for negative group identification ($\alpha$ = .61, $\omega$ = .63). The entire FHQ had good reliability ($\alpha$ = .92, $\omega$ = .92).

**Construct validity.** As with Study 2, we examined relationships between FHQ scores and continuous measures of friendship preferences and practices to explore the construct validity of the FHQ. We used identical procedures to compute the average group size and the proportion of time in groups (this time, however, we multiplied the number of interactions with two friends by 2 and the number of interactions with three friends by 3). As seen in Table 9, participants with higher FHQ scores, indicating more group-oriented friendships, reported spending time with a higher number of friends (ideal, most comfortable, and most enjoyable number of friends). This was also reflected in actual behavior as participants with higher FHQ scores spent more time in groups and socialized in larger groups.

We also examined participants' choices when selecting the number of friends they usually socialized with. Most participants ($n$ = 174, 40.56%) reported usually socializing with 3 friends at once, compared to 108 (25.17%) people reporting that they socialize with 2 friends, 77 (17.95%) usually socializing in one-to-one interactions and 70 (16.32%) reporting more interactions with 4+ friends. Participants' FHQ scores varied as a function of these choices, $F(3, 425)$ = 7.86, $p < .001$, $\eta^2 p$ = .05. Tukey post-hoc tests suggested that participants who socialized one-to-one had significantly lower FHQ scores ($M$ = 4.55, $SD$ = 1.01) than both those that socialized with three friends ($M$ = 4.92, $SD$ = .92), $p$ = .024, and those socializing with 4+ friends ($M$ = 5.24, $SD$ = .88), $p < .001$. There was no significant difference between those socializing one-to-one versus with two friends ($M$ = 4.70, $SD$ = .93) ($p$ = .73). Participants socializing with two friends also did not significantly differ from those socializing with 3 friends ($p$ = .22) but they had significantly lower FHQ scores than those that socialized with 4+ friends ($p < .001$). Finally, participants socializing with three friends did not significantly differ from those socializing with 4+ friends ($p$ = .07). An additional analysis revealed that, once again, the correlation

**Table 7. Study 3: Model fit indices and factor loadings for the four-factor model (intimacy, extraversion, positive group identification, negative group identification).**

| Model | $\chi^2$ | df | $\chi^2$/df | CFI | TLI | SRMR | RMSEA | RMSEA 90% CI |
|---|---|---|---|---|---|---|---|---|
| New 4-Factor Model | 601.60*** | 226 | 2.66 | .923 | .914 | .065 | .066 | (.060, .073) |
| Factor | | | Item | Estimate | Completely Standardized Solution | SE | p | |
| Extraversion | | | E1 | 1.22 | .90 | .06 | < .001 | |
| | | | E2 | 1.20 | .91 | .06 | < .001 | |
| | | | E3 | 1.05 | .82 | .06 | < .001 | |
| | | | E4 | 1.02 | .79 | .06 | < .001 | |
| | | | E5 | 1.20 | .85 | .06 | < .001 | |
| | | | E6 | .49 | .39 | .06 | < .001 | |
| | | | E7 | .96 | .72 | .06 | < .001 | |
| | | | E8 | .94 | .76 | .06 | < .001 | |
| Intimacy | | | I1 | 1.05 | .78 | .07 | < .001 | |
| | | | I2 | .96 | .70 | .09 | < .001 | |
| | | | I3 | .74 | .62 | .06 | < .001 | |
| | | | I4 | .98 | .81 | .06 | < .001 | |
| | | | I5 | .94 | .81 | .06 | < .001 | |
| | | | I6 | .65 | .49 | .08 | < .001 | |
| Positive Group Identification | | | GP1 | .38 | .64 | .07 | < .001 | |
| | | | GP2 | .66 | .77 | .11 | < .001 | |
| | | | GP3 | .72 | .86 | .12 | < .001 | |
| | | | GP4 | .41 | .64 | .07 | < .001 | |
| | | | GP5 | .60 | .72 | .09 | < .001 | |
| Negative Group Identification | | | GN1 | .38 | .64 | .11 | < .001 | |
| | | | GN2 | .30 | .53 | .09 | < .001 | |
| | | | GN3 | .46 | .76 | .15 | < .001 | |
| | | | GN4 | .37 | .66 | .12 | < .001 | |
| Friendship Styles | | | Extra | .85 | .65 | .10 | < .001 | |
| | | | Intimacy | .64 | .54 | .10 | < .001 | |
| | | | Positive Group I | 1.72 | .86 | .35 | < .001 | |
| | | | Negative Group I | 2.50 | .93 | .88 | < .001 | |

***p < .001.

**Table 8. Study 3: Factor correlations for the four-factor model.**

| Factor | 1 | 2 | 3 | 4 |
|---|---|---|---|---|
| 1. Extraversion | - | | | |
| 2. Intimacy | .28*** | - | | |
| 3. Positive Group Identification | .51*** | .57*** | - | |
| 4. Negative Group Identification$^\Delta$ | .67*** | .42*** | .79*** | - |

Note.

$^\Delta$s denote reverse-scored FHQ dimensions.

**Table 9. Study 3: Descriptive statistics and spearman correlations between FHQ scores and measures of friendship practices.**

| Variable | n | M | SD | Min | Max | 1 | 2 | 3 | 4 | 5 | 6 | 7 |
|---|---|---|---|---|---|---|---|---|---|---|---|---|
| 1. FHQ | 433 | 4.84 | .95 | 1.00 | 7.00 | - | | | | | | |
| 2. Ideal Number of Friends | 429 | 3.34 | 2.02 | 0.00 | 30.00 | .31*** | - | | | | | |
| 3. Usual Number of Friends | 425 | 3.47 | 3.29 | 0.00 | 42.00 | .25*** | .47*** | - | | | | |
| 4. Comfortable Number of Friends | 425 | 3.04 | 3.11 | 0.00 | 42.00 | .24*** | .57*** | .67*** | - | | | |
| 5. Enjoyable Number of Friends | 425 | 3.74 | 3.79 | 0.00 | 42.00 | .28*** | .46*** | .52*** | .61*** | - | | |
| 6. Average Group Size | 423 | 2.15 | .56 | 1.00 | 4.00 | .15** | .30*** | .53*** | .44*** | .38*** | - | |
| 7. Proportion of Time in Groups | 423 | .60 | .23 | 0.00 | 1.00 | .09 | .27*** | .46*** | .39*** | .32*** | .90*** | - |

between the FHQ scores and the theoretically unrelated participants' level of education ($M = 3.61$, $SD = 1.10$) was not statistically significant, $r_S$ (417) = .02, $p = .682$.

## Discussion

Study 3 replicated the findings of Study 2 by showing that the four-factor model of friendship styles had an acceptable fit in an additional sample. Thus, the dimensions of intimacy, extraversion, and both positive and negative group identification are relevant to friendship styles. Once again, correlations suggested that the desire for intimacy is a group-oriented trait.

Importantly, participants' FHQ scores covaried with how they categorized their socializing styles. We found positive associations between FHQ scores and self-reports of group-oriented friendship styles. Specifically, higher FHQ scores, reflecting high levels of extraversion, desire for intimacy, and group identification, significantly predicted group-oriented preferences and behaviors. Moreover, FHQ scores were higher among people who reported socializing with three or more friends at once compared to participants who tended to socialize with one or two friends at once.

In line with Study 2, correlations with FHQ scores were larger for friendship preferences than for friendship practices. While this may suggest a tendency for participants to want larger friendship groups than they socialize in, it is worth noting that the data collection for Study 3 was completed between March and April 2020, during the COVID-19 lockdown, which reduced opportunities to meet friends.

## General discussion

The present research aimed to design, refine and begin validation of the Friendship Habits Questionnaire (FHQ). The FHQ is a novel scale measuring whether a person is more likely to socialize in groups (as indicated by higher scores) or in dyads (as indicated by lower scores). We began with a theoretically-motivated model, in which individual differences in extraversion, intimacy, competitiveness, and group identification were expected to predict FHQ scores. Study 1 found that, compared to the theoretically motivated four-factor model, the structure of the FHQ was best described with a six-component model including extraversion, intimacy, competitiveness as two separate dimensions (enjoyment of competition and contentiousness) and group identification as two separate dimensions (positive and negative group identification). Study 2 further explored the structure of the FHQ and found that a four-factor model excluding the two dimensions of competitiveness (enjoyment of competition and contentiousness) had a better fit compared to the six-factor model that emerged from Study 1 and to the theoretically motivated four-factor model. Study 3 reevaluated the fit of the four-factor model in a separate sample and confirmed adequate fit. Moreover, Studies 2 and 3 showed that FHQ

had good reliability and that high FHQ scores were consistently associated with self-reports of socializing in larger friendship groups and enjoying group interactions.

Overall, these results support our prediction that intimacy, group identification, and extraversion are relevant to friendship group size. These dimensions emerged in all three assessed models with the only changes relating to the structure of group identification (divided into the positive and negative component). This change was theoretically justified given that existing group identification scales [28, 29] include subscales roughly mapping onto positive and negative aspects of group identification similar to those found in our study. Finally, the FHQ dimensions, in particular intimacy, were included in an existing measure of ideal friendship standards [56] and may be relevant to social bonds across all ages of the human lifespan [7, 8, 31, 33]. Competitiveness was only weakly associated with other dimensions and was discarded from the final model. This finding is consistent with the results obtained by Hall [56]. In this previous study, which explored the ideal standards of friendships, competitiveness was linked with the resource-based aspects of friendships, including also friends' attractiveness, wealth, or business connections. However, items measuring competitiveness were later removed due to poor factor loadings. This finding, alongside the current results, suggests that competitiveness is less important to friendship than other factors. The low fit of competitiveness with other factors also dovetails with findings by Cheng and Chan [57], who did not find significant associations between competitiveness and intimacy, despite predicting a negative relationship. Consistently, we did not observe significant relationships between these two dimensions (see Table 3 and S2 Table). Intimacy was significantly and positively correlated with group identification, highlighting the need for future studies examining self-disclosure in groups and in particular friendship groups [54].

Studies 2 and 3 tested the validity of the FHQ by examining the relationships between participants' FHQ scores and their self-reported friendship preferences and behaviors in general and over the last 3 months. In both studies we found that FHQ scores significantly covaried with both continuous and categorical measures such that higher scores indicated socializing with larger groups of people or seeking such interactions.

To our knowledge, this is the first study assessing multiple traits to measure one's socializing style in terms of group size. Existing studies tended to focus on one or two traits of group- versus dyadic-oriented people at a time. Here we find that individual differences in a self-report, trait-based questionnaire significantly predict variations in socializing preferences and behaviors. Thus, the FHQ can be used to measure friendship styles in a more subtle way that does not require directly asking participants about their real-world socializing behaviors. This is important because assessing the size of groups in which participants socialize might not always be easy or even reliable. For example, in our studies people's self-reported estimates of the number of friends usually present in social interactions were only moderately correlated with the average group size calculated from participants' reports of their total interactions in the last 3 months. This suggests estimates are not always accurate. The period that participants are asked about may also reduce validity. Measures of social interactions that rely on retrospective memory for events that occurred over long periods may be subject to more recall error. On the other hand measuring social interactions in real-time, for example with experience sampling methods, may lead to greater dropout rates or simply be unfeasible. In contrast, too short a periods might not be reflective of one's long-term socializing styles. For example, students in exam period or working adults with looming deadlines might see their friends less, whereas around national holidays people might see their friends more. Using a trait-based approach like the FHQ avoids these problems as the questionnaire asks the person to think about their general behavior with friends and, when scored, gives an indication of the likelihood that this person will socialize in groups. This new questionnaire adds to the existing

measures of friendship standards [56] or qualities of specific friendships [e.g., 13]. Examining group versus dyadic-oriented friendship styles is a promising possibility for future research. There is, for example, evidence that friendship group sizes differ across countries [e.g., 58] and FHQ allows to explore cross-cultural variation in friendship interactions and traits associated with group versus dyadic-oriented friendship styles. Additionally, researching people's tendency to socialize in larger groups or dyads can shed more light on cultural variation in other behaviors or competencies, such as emotion recognition [e.g., 59] or expression [60].

One possible limitation of the current studies is their reliance on self-reports, potentially prone to social desirability bias and inaccurate memories, especially for reporting social interactions over the last 3 months. In addition, participants' reports of their social interactions in the last 3 months for Study 3 might have been affected by the Covid-19 as both lockdown in the UK and the switch to online learning for educational organizations in Sweden started at the end of March 2020. During the lockdown, social interactions, particularly group ones, might not have been as common as before the pandemic. However, it is worth acknowledging that, as data collection for Study 3 was finished at the end of April 2020, asking participants about their socializing in the last 3 months should have provided insights into at least 2 months' worth of interactions unaffected by restrictions. Nonetheless, future research should explore the relationships between the FHQ scores with other types of measures of socializing, including diaries or experience sampling. It is also important to examine FHQ in experimental or longitudinal designs since the cross-sectional nature of the present research does not allow causal conclusions.

Another limitation of our study is that the dimensions studied as relevant to friendship styles were selected by the researchers involved. Even though this selection was guided by a theoretical perspective based on existing research, we cannot rule out the possibility that our literature search missed other potentially relevant group- and dyad-oriented traits. An alternative approach to Study 1 would have been to present participants with a longer and more labor-intensive questionnaire with items indexing a vast range of friendship-related dimensions, and to use a item-reduction approach, much like Hall's questionnaire on ideal expectations of friends [56]. Here we followed the approach suggested by Hinkin and Tracey [42] who argue that a theoretically-driven approach is as effective as the more time-intensive item-reduction. Indeed, the FHQ reached the highest level of reliability in Studies 2 and 3 and achieved an acceptable fit, suggesting that the questionnaire is fit to measure group-oriented friendship. Nonetheless, future research should consider whether the FHQ would benefit from the inclusion of other traits. There might also be issues with our samples. The average participant age in both Study 2 and Study 3 was between 20–30 years old which might not be reflective of children, adolescents and older age groups. In addition, our samples included slightly more females (55% in Study 2; 61% in Study 3) than males and gender has been shown to influence group size [6–8]. Finally, participants only filled out the survey once, making it impossible to estimate the test-retest reliability.

Some people usually socialize in friendship groups and others spend more time in dyadic interactions. Here we propose a new measure of such tendencies based on personality traits and individual characteristics and demonstrate that variations in this measure predict both friendship preferences and behaviors.

## Supporting information

**S1 Fig. Study 1: Scree plot showing eigenvalues of each principal component for the oblimin rotation.**
(DOCX)

**S2 Fig. Study 1: Scree plot showing eigenvalues of each principal component for the varimax rotation.**
(DOCX)

**S1 Table. Study 1: Descriptive statistics and component loadings for the PCA analysis with varimax rotation.**
(DOCX)

**S2 Table. Study 2: Factor loadings the four-factor theoretical model (extraversion, competitiveness, intimacy and group identification).**
(DOCX)

**S3 Table. Study 2: Factor correlations for the four-factor theoretical model.**
(DOCX)

**S4 Table. Study 2: Factor loadings the six-factor model (extraversion, enjoyment of competition, contentiousness, intimacy, positive group identification and negative group identification).**
(DOCX)

## Acknowledgments

Thank you to Conall McGuinness for his overall support and intriguing conversations around friendship. He is a dearly missed friend who we will never forget. Thanks to Marion Karlsson Faudot and Lina El Manira for their help with data collection and their continued enthusiasm and friendship.

## Author Contributions

**Conceptualization:** Philip Howlett, Tomas Jungert.

**Data curation:** Philip Howlett.

**Formal analysis:** Philip Howlett, Gülseli Baysu, Tomas Jungert.

**Investigation:** Philip Howlett, Tomas Jungert.

**Methodology:** Philip Howlett, Tomas Jungert.

**Project administration:** Philip Howlett, Tomas Jungert.

**Resources:** Philip Howlett.

**Supervision:** Gülseli Baysu, Anthony P. Atkinson, Tomas Jungert, Magdalena Rychlowska.

**Validation:** Philip Howlett.

**Writing – original draft:** Philip Howlett, Tomas Jungert, Magdalena Rychlowska.

**Writing – review & editing:** Philip Howlett, Gülseli Baysu, Anthony P. Atkinson, Tomas Jungert, Magdalena Rychlowska.

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
