## [Decision Letter · Decision Letter 0]

24 Nov 2022

PONE-D-22-25565Friendship Habits Questionnaire: A measure of group- versus dyadic-oriented socializing preferencesPLOS ONE

Dear Dr. Howlett,

Thank you for submitting your manuscript to PLOS ONE. After careful consideration, we feel that it has merit but does not fully meet PLOS ONE’s publication criteria as it currently stands. Therefore, we invite you to submit a revised version of the manuscript that addresses the points raised during the review process.

We look forward to receiving your revised manuscript.

Kind regards,

Fang Wang

Academic Editor

PLOS ONE

Journal Requirements:

Reviewers' comments:

Reviewer's Responses to Questions

**Comments to the Author**

1. Is the manuscript technically sound, and do the data support the conclusions?

Reviewer #1: Yes

Reviewer #2: Yes

Reviewer #3: Yes

2. Has the statistical analysis been performed appropriately and rigorously? 

Reviewer #1: Yes

Reviewer #2: Yes

Reviewer #3: Yes

3. Have the authors made all data underlying the findings in their manuscript fully available?

Reviewer #1: Yes

Reviewer #2: Yes

Reviewer #3: Yes

4. Is the manuscript presented in an intelligible fashion and written in standard English?

Reviewer #1: Yes

Reviewer #2: Yes

Reviewer #3: Yes

5. Review Comments to the Author

Reviewer #1: The authors report three studies that examined individual differences associated with socializing in pairs versus groups and put forward the Friendship Habits Questionnaire (FHQ) for measuring socializing preferences (either dyad oriented or group oriented).

Study 1 explored the dimensions of the FHQ using principal component analysis and expected four dimensions to appear: extraversion, group identification, intimacy, and competitiveness. The items from each dimension were taken from existing questionnaires and adapted to the study’s purposes (i.e., modified to describe friendship groups). Results indicated that instead of the predicted four dimensions, the analysis revealed 8 components of which two were not interpretable resulting in six components describing the dimensionality of the FHQ: extraversion, intimacy, competitiveness, positive group identification, negative group identification, and contentiousness. A weakness of this study is the small sample size (N = 49).

Study 2 asked 353 participants to rate their own friendship habits using the FHQ and examined its structure using confirmatory factor analysis. The authors compared the theoretical four-factor model with the empirical six-factor model derived from Study 1. Finally, to explore the construct validity of the FHQ, obtained scores were correlated with self-report measures of friendship size preferences and recent friendship group size experiences. Results yielded a poor fit for both models. Taking the results of the first set of analyses into account, a modified four-factor model was proposed and analyzed in the same study comprising the factors intimacy, extraversion, positive group identification and negative group identification. That is, competitiveness and contentiousness were dropped from the six-factor model. The latter model showed acceptable fit and was used for additional analyses. Supporting evidence was found for the FHQ’s construct validity.

Study 3 replicated the fit of the modified four-factor model derived from Study 2 using 433 participants. As in Study 2, participants answered questions about their friendship group size preferences and their friendship group size practices. Results showed an acceptable fit of the model with the exception of the relative chi-square index and the RMSEA. Again, supporting evidence was found for the construct validity of the FHQ.

Overall, the paper is clearly written and the research seems well executed. The data are interesting and the FHQ can be used in future research projects. Another strength of the present manuscript is that it has strong focus. While there are different sidetracks available (gender differences, group size analyses), the authors stay on course pursuing their main objective. As indicated, the small sample size of Study 1 is a weakness, especially because it is a short study to run. After addressing minor concerns, I would consider it publication-worthy.

Major issues: None.

Minor comments:

(1) Page 4. Lines 65–68: “Here we present the Friendship Habits Questionnaire (FHQ), a new tool measuring whether one’s socializing preferences are more dyadic versus group-oriented based on individual differences in needs for intimacy, competitiveness, group identification, and extraversion.”

Comment: The order of individual difference variables is listed as needs for intimacy, competitiveness, group identification and extraversion and subsequently described in more detail in the following paragraphs in the order extraversion, group identification, intimacy, competitiveness.

The authors may consider bringing these into harmony by changing the order in lines 65–68 to correspond with the order of subsequent elaboration.

(2) Page 5. Lines 88–89. The sentence beginning with the words “Another relevant construct is

intimacy, (…)” may start a new paragraph just like the other individual difference variables, instead of being combined into a single paragraph with group identification.

(3) Page 8. Lines 172–175. “We used varimax rotation following the procedure used by Hinkin and Tracey (41), given that we expected correlated factors, we also ran a PCA using an oblimin rotation, which showed an identical pattern of six interpretable components.”

Comment: Varimax rotation is an orthogonal type of rotation. In orthogonal rotation, the factors are constrained to be uncorrelated, that is, one believes the underlying factors are independent. The use of orthogonal rotation may produce misleading solutions where the factors are expected to be intercorrelated, for example in the present study, a questionnaire whose latent structure entails several interrelated dimensions of a broader construct. Fortunately, the authors have also run a PCA with oblimin rotation which yielded similar results. However, the above reference to and use of varimax should be addressed to avoid confusion.

(4) Page 10. Table 1. The second intimacy item has a notification “Error! Bookmark not defined.” that should be removed if present in the submitted manuscript.

(5) Page 16. Lines 310–311. “As shown in Table 4, this new model showed an acceptable fit on all indices except the relative chi-square, and was retained for further analyses.”

Comment: The relative chi-square is 1.88 in Table 4, and hence fits the criterion for acceptable model fit stated on page 15 line 285: “χ²/df < 2.0”.

(6) Page 21–22. Lines 386—389. “Although a previous study (32) showed that self-disclosure occurs more in dyads than in larger groups, this research focused on interactions between strangers, and it is unclear how self-disclosure would play out among friends.”

Comment: There may be some room for misunderstanding if “this research” is read as the present research or the present paper instead of the research from reference (32), which did focus on strangers instead of friendship interactions.

(7) Page 23. Lines 428–429. “We assessed the goodness of fit of the four-factor second-order model using the same criteria as in Study 2.”

Comment: In Table 7, on page 23, the SRMR and RMSEA values are missing.

(8) Page 27. Lines 477–478. “Study 3 replicated the findings of Study 2 by showing that the four-factor model of friendship preferences had a good model fit in an additional sample.”

Comment: On Page 23 lines 429–431 describing the results of Study 3 it states: “The model showed an acceptable fit with the exception of the relative chi-square and RMSEA (see Tables 7 and 8 for factor loadings and correlations).”

Therefore, it would not be accurate to describe the model fit as “good” in the Discussion section on page 27.

(9) Page 29. Lines 515–517. “Finally, the FHQ dimensions, in particular intimacy, were included in an existing measure of ideal friendship standards (56) and may be relevant to social bonds formed by primate species (34).”

Comment: The focus of the paper is on the scale development of the FHQ questionnaire for measuring socializing preferences in humans. The reference to primate species appears to detract the focus away from the discussion of the main findings.

Reviewer #2: Thanks for inviting me to review the manuscript “Friendship Habits Questionnaire: A measure of group- versus dyadic-oriented socializing preferences”. The purpose of the manuscript is to present the process of development of a new questionnaire to assess friendship habits with the focus on preferences of amount of persons friendship is experiences with. The authors call it “group versus dyadic- oriented”. The manuscript contains three sequential studies which describes the refinement of the underlying constructs and first psychometric properties to validate the new measure FHQ (Friendship Habit Questionnaire) such as construct validity or reliability.

The overall flow of the paper is solid and organized in a fluent and scientific manner and thus informative. The paper is in general quite long, as one could expect in describing three validation studies. As friendship is such an important part of life, a variation of assessments with distinct underlying concepts are needed to understand this complex phenome better. Therefor the authors can be complemented for their efforts and hard work to present the development of the FHQ.

Despite the accolades, there are several suggestions to improve the strength of the manuscript.

Abstract:

It should be explicit that this is a new measure and that the three performed studies investigated psychometric properties and refinement of the underlying construct.

It is further confusing to first read about competitiveness and later in the results about other dimensions. It might be more important to present the overall aim of all three studies.

Introduction:

Please explain more in detail why this new measure is needed (line 61- 65 is too short)

Line 68: it is definitely needed to explain how you came to the construct with the four dimensions you started with and made factor analysis.

Please insert a clear overall research question (Line 109) instead of describing the overall examination.

Line 110. Which previous findings? Please insert again references.

Line 116-124: Instead of describing the organization of the three studies, I would prefer a figure (type flowchart? ) to support readers orientation.

Study 1

Please organize each study (1-3) along research questions, hypothesis followed (as you did) by methods (participants, recruitment, material) and analysis. This would make it much easier to retrain from repetition such as in line 128 – 131.

I propose to put the description of the FHQ with the original underlying concepts derived from other measures in the introduction (line 150 – 168). Then each of the sequential studies are shorter and easier to understand.

Line 170 – 174: differentiate between analysis and results instead of mixing them. This is for me as reader confusing.

Table 1 is informative, but the Asterisks confusing with p>0.5. So I suggest using another graphical solution to indicate reverse-scores FHQ items.

Discussion Study 1: very informative

Study 2

See suggestion of organization of study 1.

Line 242 -245 redundancy, instead make the aims and /or the different research questions of study 2 explicit (may be also possible to put into the flowchart/figure?)

Line 267: Again, separate analysis and results.

Table 4: I miss explanation of the Friendship preferences. Are these summary scores?

Table 6: the first line does not fit with the following ones. Please adjust.

Study 3

Line 412 and 416: give a methodological reason for the attention check.

Line 412: please explain why you changed the options to describe the preference of friendship habits as compared to study 2.

Line 425: I suggest deleting the emotion recognition task if not explained.

General Discussion

I disagree with the main aim to validate the FHQ. To my understanding, the three sequential studies develop the FHQ by examining and adapting the construct to achieve best validity and a small part of reliability. As validation of a measurement are somehow much more than you did (e.g. concurrent validity, age and gender validity….) , I suggest to stay here modest.

Line 540: I expect here more suggestions how to use the FHQ and practical implications. Asking directly are 4 main questions, while the FHQ has 30! Line 544 What prosper utility see the authors?

Line 550: I think you should reflect on more limitations of your work. One is the participants of these studies: first their age rang is quite small and secondly the gender inequality may have an influence on your results. Please also reflect on the influence the Covid lockdown might have on the results. Another limitation is that reliability is not measured with test-retest reliability or stability over time.

Again, the authors should be commended on their time and effort develop a new measurement I the field of friendship evaluation. Continued revisions will strengthen the summaries of this early stage manuscript document.

Reviewer #3: This research aimed to validate a measure of friendship habits--particularly one's preference for socializing in groups versus dyads. The authors showed that the structure of Friendship Habits was best described with four dimensions, namely, extraversion, intimacy, positive group identification, and negative group identification. They also established the reliability and construct validity of the measure.

The manuscript is well-written and carefully provides the results of the statistical analyses that support the authors' conclusion. I only have a couple of suggestions.

First, it would be great if the authors are clearer and more consistent with what the FHQ score actually means, throughout the manuscript. For example, the title of the manuscript says "a measure of group-oriented versus dyadic-oriented socializing preferences." In Genenral Discussion, however, it says "a novel scale measuring whether a person socializes in groups."

Relatedly, I found it unclear whether the FHQ score indicates the extent to which one prefers socializing in groups (of three people or more) or the group size that a person prefers in socialization.

Second, the authors mentioned that Study 1 provided the participants with the definition of friendship habits. Importantly, the definition of FH pretty much reflects the four theory-driven factors including the factor "competitiveness". It would be helpful if the authors could explain why they provided such a definition to the participants, and how it might have affected the participants' responses.

Third, the authors mentioned that the participants in Study 3 were provided with the descriptions of friendship interactions and example activities. I recommend the authors to explain what the descriptions were, even if it is quite brief, as it is important how the term 'friendship' was defined in this research.

6. PLOS authors have the option to publish the peer review history of their article (what does this mean?). If published, this will include your full peer review and any attached files.

Reviewer #1: No

Reviewer #2: No

Reviewer #3: No

---

## [Author Response · Author response to Decision Letter 0]

24 Jan 2023

Please see attached document titled "Response to Reviewers" in this resubmission application for responses to all reveiwer and editor comments

---

## [Decision Letter · Decision Letter 1]

2 May 2023

Friendship Habits Questionnaire: A measure of group- versus dyadic-oriented socializing styles

PONE-D-22-25565R1

Dear Dr. Howlett,

We’re pleased to inform you that your manuscript has been judged scientifically suitable for publication and will be formally accepted for publication once it meets all outstanding technical requirements.

Kind regards,

Fang Wang

Academic Editor

PLOS ONE

Additional Editor Comments (optional):

Reviewers' comments:

Reviewer's Responses to Questions

**Comments to the Author**

1. If the authors have adequately addressed your comments raised in a previous round of review and you feel that this manuscript is now acceptable for publication, you may indicate that here to bypass the “Comments to the Author” section, enter your conflict of interest statement in the “Confidential to Editor” section, and submit your "Accept" recommendation.

Reviewer #1: All comments have been addressed

Reviewer #2: All comments have been addressed

2. Is the manuscript technically sound, and do the data support the conclusions?

Reviewer #1: Yes

Reviewer #2: Yes

3. Has the statistical analysis been performed appropriately and rigorously? 

Reviewer #1: Yes

Reviewer #2: Yes

4. Have the authors made all data underlying the findings in their manuscript fully available?

Reviewer #1: Yes

Reviewer #2: No

5. Is the manuscript presented in an intelligible fashion and written in standard English?

Reviewer #1: Yes

Reviewer #2: Yes

6. Review Comments to the Author

Reviewer #1: (No Response)

Reviewer #2: Congratulation for the performed changes.The manuscript improved a lot. Good luck with using theis questionnaire further.

7. PLOS authors have the option to publish the peer review history of their article (what does this mean?). If published, this will include your full peer review and any attached files.

Reviewer #1: No

Reviewer #2: **Yes: **Dr. Beate Krieger

---

## [Editor Report · Acceptance letter]

2 Jun 2023

PONE-D-22-25565R1 

Friendship Habits Questionnaire: A measure of group- versus dyadic-oriented socializing styles 

Dear Dr. Howlett:

I'm pleased to inform you that your manuscript has been deemed suitable for publication in PLOS ONE. Congratulations! Your manuscript is now with our production department. 

Kind regards, 

on behalf of

Dr. Fang Wang 

Academic Editor

PLOS ONE